# Foliar Fertilization of Crop Plants in Polish Agriculture

**Rafał Januszkiewicz** [1], **Grzegorz Kulczycki** [1,*] and **Mateusz Samoraj** [2]

[1] Institute of Soil Science Plant Nutrition and Environmental Protection, Wrocław University of Environmental and Life Sciences, Grunwaldzka Str. 53, 50-363 Wrocław, Poland; rafal.januszkiewicz@upwr.edu.pl

[2] Department of Advanced Material Technologies, Faculty of Chemistry, Wrocław University of Science and Technology, Smoluchowskiego 25, 50-372 Wrocław, Poland; mateusz.samoraj@pwr.edu.pl

\* Correspondence: grzegorz.kulczycki@upwr.edu.pl; Tel.: +48-71-320-5654

**Abstract:** Foliar fertilization makes it possible to quickly supply plants with deficient nutrients, in case of both their deficiency in the soil and hindered uptake. Crops are characterized by varying nutrient requirements for micronutrients, as well as varying sensitivity to their deficiency. The paper presents practical aspects of the foliar feeding of plants with micronutrients using foliar fertilizers, and their general classification and characteristics. The requirements of basic crops (cereals, rapeseed and corn) for the application of micronutrient fertilization and the degree of their sensitivity to micronutrient deficiency were characterized. The market of foliar fertilizers and the directions of its development were evaluated. The possibilities of foliar fertilizers containing amino acids and silicon, and the possibility of using them for biofortification are presented. It was found that foliar fertilization is one of the most popular and effective methods for the biofortification of plants, as it allows the delivery of the right amount of specific elements in a specific stage of plant development and is thus an economical and environmentally safe procedure. In conclusion, the analysis of the foliar fertilizer market shows that its development is very dynamic, and foliar fertilization is becoming one of the basic elements in effective crop production. Further expansion of the range of foliar fertilizers produced should be expected in accordance with the growing expectations of agricultural producers.

**Keywords:** foliar fertilization; micronutrients; biofortification; cereals; corn; rapeseed

## 1. Introduction

The systematic growth of the world's population requires providing it with sufficient food [1]. With the shrinking in areas devoted to crop cultivation, the only possible method for meeting the nutritional needs of such a large population is to increase the yield per unit area [2]. This increase is associated with the depletion of nutrients contained in soils, by way of uptake by plants and dispersal in the environment, resulting in severe nutrient deficiencies. The way to make up for these deficiencies is to use fertilizers, especially mineral fertilizers, in which the components are most often found in readily available forms. The use of mineral fertilizers in a soil-based manner is linked to their transformation in the soil, which affects their availability, resulting in the excessive or insufficient uptake of nutrients by plants. These phenomena can affect the quantity and quality of the yield obtained and the loss of nutrients due to their leaching into the soil and their excessive volatilization into the atmosphere, causing increasing environmental pollution [3].

The aforementioned aspects and problems associated with the use of mineral fertilizers inspire the search for global solutions to increase the efficiency of the use of nutrients from fertilizers, enabling the production of sufficient quantities of good-quality food while maintaining a healthy environment. These goals are served by, among others, the UN Sustainable Development Strategy, which plans to eliminate hunger and all forms of malnutrition worldwide by 2030 [4], as well as the European Green Deal strategy [5], which additionally indicates that by 2030, nutrient losses should be reduced by at least 50% and fertilizer use should be reduced by a minimum of 20%.

Achieving these goals is possible, but it means developing more efficient and effective fertilizer systems. Their goal should be to increase agricultural productivity with less fertilizer inputs [6]. Of the fertilization methods currently in use, such expectations are largely met by the foliar delivery of nutrients, the effectiveness of which can be up to several times greater than that of soil application [7].

The delivery of nutrients with foliar fertilizers generally involves (1) the application of an aqueous solution to the plant surface in the form of droplets, (2) their retention on the leaf surface, (3) the absorption of the nutrient into living plant cells and (4) the translocation of the nutrient to areas where it can be used by the plant in its life processes. The efficiency of foliar fertilization largely depends on the absorption mechanisms of foliar-applied molecules, as they are affected by many environmental factors. There are review papers in the literature that focus on the mechanisms of penetration of foliar-applied nutrient solutions through the leaf surface [8–10].

Foliar fertilization is an increasingly common way in plant nutrition to provide nutrients to plants, and nowadays, it is also aimed at the following goals [10]: (1) increasing the concentration of components in plant foods with biofortification (enrichment), using fertilizers containing readily available forms of deficient elements (selenium, iodine, zinc, iron), the deficiency of which occurs in the diet of animals and humans and affects about 25% of the population; (2) improving the utilization of the supplied elements by introducing substances that increase their uptake, water uptake and resistance to stress caused by abiotic factors. These include biostimulants that activate plant metabolic processes, reduce stress and alleviate nutrient deficiencies. These are substances that are said to completely change crop production in the near future.

The growing demand for quality food products, resulting from a growing world population, requires the use of optimal solutions to provide plants with optimal conditions for their growth and development [11]. While the weather conditions that affect the course of vegetation and crop yields are difficult to predict and control [12], in the process of feeding plants, we have the opportunity to take measures to increase the availability of nutrients for plants.

Proper foliar fertilization is one of the indispensable elements of agrotechnology to achieve high yields. This method of fertilization makes it possible to provide plants with all the necessary nutrients in every stage of development. It is also an effective way to stimulate and realize the potential of a given plant. It is an excellent way to support the root system of plants during periods of intensive growth. It makes it possible to provide plants, in a quick and effective way, with missing components during periodic shortages, resulting from the inability to absorb these components from the soil. Through foliar fertilization, we can also selectively supply micronutrients to sensitive plant species, so we are able to optimally meet the nutritional needs of plants. Foliar fertilizers provide high efficiency of fertilization and thus allow us to reduce the amount of components we introduce into the environment.

It is assumed that foliar fertilizers, depending on the crop, the level of agrotechnique, soil quality, weather conditions and, very importantly, the quality of the products, give the possibility of obtaining a yield increase of several to several tens of percentage points [13–17]. By supplying the plant with the missing essential nutrients, we can influence the yield and its quality parameters, which was stated as early as 1972, more than 50 years ago [18]. Thanks to foliar fertilization, we can also enrich plants in selected mineral components [19].

The market of foliar fertilizers in Poland is growing dynamically, gaining importance. It is estimated that more than 2000 different fertilizer products are currently available, and there is a 10-fold increase in their quantity compared with 2009 [7]. The wide range of foliar fertilizers on the market allows agricultural producers to choose products tailored to their current needs and field situation. However, the effective use of these fertilizers requires expert knowledge of the possibilities and advisability of their application. Agricultural producers expect products that are simple to use, characterized by full solubility, easy to dose and well miscible with chemicals, if such mixtures are approved for use [20].

Nowadays, in addition to the high nutrient content of foliar fertilizers, fertilizer manufacturers enrich their composition with various substances of a stimulating nature or supporting plant metabolism (plant extracts, algae, vitamins or amino acids) [21–23]. This makes it possible to increase the plant's resistance to stress factors, diseases and pests, or to support plant regeneration after a stress factor. Preininger et al. [24] emphasize the positive effect of using bacteria, fungi or viruses on foliar fertilization of plants.

Currently, the rules for the introduction of foliar fertilizers to the Polish market are regulated by Polish regulations [25] and EU regulations on, inter alia, making fertilizer products available on the EU market [26] According to the regulations, in order for a fertilizer to be approved for marketing, it must be made from raw materials that meet the requirements of one of the component material categories (CMCs) in Annex 2 [26]. So, among other things, what is important is the ingredient itself, its origin or its purity, while it is not important, for example, the particle size or chemical form of the element in question. In practice, most manufacturers aim to introduce and make their solutions available in accordance with the EU regulations (mainly due to the freedom of circulation in member countries), and if this is not possible, national regulations are used.

## 2. Discussion

### 2.1. Nutritional Demands of Cereals, Rapeseed and Corn

The nutritional requirements of cultivated crop species vary widely, and these needs are further influenced by the direction of production, habitat conditions, and weather. Significant differences in nutritional requirements also exist for varieties within a single species [27]. By using proper fertilization, we can also influence quality parameters, such as protein quality and content, gluten or sedimentation rate [28,29]. Therefore, it is very important that the size of fertilizer doses is adapted to the nutritional requirements of the plant species grown and the selected variety and that the current abundance of soils is taken into account, so that fertilization, which is intended to improve quantitative and qualitative parameters, does not have the opposite effect, i.e., a decrease in yield [30].

The contents of the forms of available nutrients in soils vary widely and depend on the species of soil, among other factors. Heavier soils, which contain more clay fractions, are characterized by a higher content of micronutrients compared with lighter, sandy soils [31]. Nutrient compactness can also be affected by tillage intensity, crop rotation and applied fertilization. In planning nutrient rates, it is important to know the content of nutrients in the soil in soluble (available) forms for plants. An example is the determination of plant doses of phosphorus, as it may turn out that despite the high content of the total form of this nutrient in the soil, up to 80% of it may be in a form that is not available to plants [32].

On the basis of long-term studies in Poland carried out by chemical and agricultural stations in cooperation with the Institute of Crop and Soil Sciences (IUNG) [31], a large share of soils with low abundance of micronutrients was found, especially for boron and copper (Table 1).

**Table 1.** Percentage of soils low in available forms of micronutrients in Poland.

| Microelement | 1987–1993 | 1994–1999 | 2000–2012 | 2016–2017 | |
| --- | --- | --- | --- | --- | --- |
| | | | | Wheat | Rapeseed |
| Boron (B) | 75 | 79 | 74 | 19 | 45 |
| Copper (Cu) | 37 | 34 | 34 | 30 | 14 |
| Iron (Fe) | - | 28 | 21 | 21 | 10 |
| Manganese (Mn) | 11 | 7 | 3 | 16 | 15 |
| Molybdenum (Mo) | 23 | - | - | - | - |
| Zinc (Zn) | 14 | 13 | 17 | 20 | 14 |

- data not available.

Soil pH, which strongly influences the effectiveness of agrotechnical treatments, also has an impact on limiting the availability of components [33]. In 2016, about 72% of soils in Poland were characterized by very acidic or acidic pH (41% very acidic soils, 31% acidic soils); 19%, slightly acidic pH; and 9%, neutral pH [34]. Low soil pH contributes to an increase in the toxicity of aluminum ions to the root system, a reduction in the development of beneficial microorganisms, or the hydration and leaching of nutrients deep into the soil profile.

Fertilization, both in soil and foliar forms, should ensure that plants have access to nutrients in an amount that covers their nutritional requirements, especially in critical stages of their development. Stanislawska-Glubiak and Korzeniowska [31] indicate that crop plants differ significantly in their sensitivity to micronutrient deficiency (Table 2).

**Table 2.** Sensitivity of crop plants to micronutrient deficiencies.

| Crop Plant | B | Cu | Mn | Mo | Zn |
|:---:|:---:|:---:|:---:|:---:|:---:|
| Wheat | 1 | 3 | 3 | 0 | 1 |
| Barley | 0 | 3 | 2 | 1 | 0 |
| Rye | 0 | 0 | 2 | 0 | 0 |
| Triticale | 0 | 1 | 1 | 0 | 0 |
| Oats | 0 | 3 | 3 | 1 | 0 |
| Rape | 3 | 1 | 2 | 2 | 0 |
| Sugar beet | 3 | 2 | 3 | 2 | 1 |
| Corn | 2 | 2 | 2 | 0 | 3 |

0—none; 1—small; 2—medium; 3—large.

Among the basic nutrients, six macronutrients and six micronutrients can be specifically distinguished in terms of their indispensability to crop plants. From among the basic macronutrients such as nitrogen, phosphorus, potassium, magnesium, sulphur and calcium, plants take up the most nitrogen and potassium and this can be as much as over 200 kg per hectare. On the other hand, among the micronutrients boron, copper, iron, manganese, molybdenum and zinc, there is a wide variation in their requirements for plants. These needs range from a few to several hundred grams per hectare. It clearly follows that the main source of macronutrients for plants must be soil fertilization, while in the case of micronutrients, only foliar application can fully meet the nutritional needs of the plant [7].

*2.2. Cereals*

Cereals are among the most popular crops in Poland and also dominate global production. In Poland, in 2021, the area sown with cereals was 7.45 million hectares [35], and the popularity of this group of crops is due to the possibility of their versatile use. Cereal grains are used in the food and feed industries but can also be used in the energy or pharmaceutical industries [35]. The area of cereal crops in Poland has fluctuated over the past 50 years. The 1965–1980 period saw a decline in the area under cultivation, followed by an increase in interest in this group of crops between 1980 and 2000, and another decline since 2001. The 2000 season saw the largest area of cereal planting in Poland—8.81 million hectares [36].

Cereal cultivation is dominated by winter varieties, which have higher yields and better economic efficiency of production (Table 3). The estimated average yield of winter cereals in Poland in the 2021 season was 46 dt ha$^{-1}$, while that of spring cereals was 35 dt ha$^{-1}$ [34].

**Table 3.** Cereal yield in Poland in 2010–2021.

| Species of Cereal | 2010 | 2015 | 2016 | 2017 | 2018 | 2019 | 2020 | 2021 |
|---|---|---|---|---|---|---|---|---|
| | dt ha$^{-1}$ | | | | | | | |
| **Basic Cereals with Cereal Mixtures** | **35.1** | **36.7** | **37.5** | **40.0** | **32.3** | **35.2** | **44.8** | **42.6** |
| Winter wheat | 45.7 | 47.6 | 47.2 | 51.1 | 43.0 | 46.4 | 54.2 | 51.8 |
| Spring wheat | 34.3 | 33.5 | 38.3 | 38.5 | 31.5 | 32.6 | 41.7 | 39.6 |
| Rye | 26.9 | 27.8 | 28.9 | 30.6 | 24.2 | 27.2 | 35.1 | 33.1 |
| Winter barley | 40.7 | 41.3 | 44.6 | 47.1 | 37.8 | 43.0 | 51.1 | 47.7 |
| Spring barley | 33.0 | 33.0 | 35.8 | 38.0 | 29.5 | 32.1 | 40.0 | 37.8 |
| Oats | 26.4 | 26.5 | 28.4 | 29.8 | 23.5 | 24.9 | 33.2 | 31.4 |
| Winter triticale | 35.2 | 36.3 | 37.1 | 40.4 | 32.8 | 35.9 | 45.0 | 43.1 |
| Spring triticale | 28.4 | 28.4 | 31.7 | 32.9 | 25.1 | 27.5 | 36.4 | 33.7 |
| Winter cereal mixes | 30.9 | 30.9 | 32.4 | 34.4 | 28.2 | 30.6 | 38.1 | 36.6 |
| Spring cereal mixes | 30.5 | 27.2 | 29.8 | 32.2 | 25.0 | 26.2 | 34.5 | 33.7 |

Different cereal species are characterized by different nutrient requirements. Winter wheat requires 22–26 kg of nitrogen (N), 8 kg of phosphorus ($P_2O_5$), 5 kg of potassium ($K_2O$), 2 kg of magnesium (MgO) and 1 kg of calcium (CaO) to produce 1 t of grain. In turn, this species' micronutrient requirements in grams per hectare are as follows: boron (B), 115 g; copper (Cu), 120 g; manganese (Mn), 500 g; molybdenum (Mo), 7 g; and zinc (Zn), 350 g [28].

*2.3. Rapeseed*

In recent years, there has been a steady increase in interest in the cultivation of rapeseed. In the 2021 season, the area under cultivation of this crop in Poland amounted to 0.99 million hectares [35]. In comparison, in 1947, the cultivation area was only 61 thousand hectares, and in 2002, 439 thousand hectares. With the increase in the area of cultivation of this plant, an increase in the yield of production per 1 ha is also noted. Over the past 10 years, the average yield per 1 ha has increased from 23.6 dt to 32.1 dt ha$^{-1}$, or as much as 36%. The increase in the yield of this crop is due to the introduction of new, improved varieties and improved agrotechnology, including fertilization. The high popularity of rapeseed, as in the case of cereals, is the result of significant market demand for this type of product. Among other uses, rapeseed is used in the production of cooking oils and in the production of feedstuffs, as well as biofuels [37].

Rapeseed is characterized by very high nutrient requirements. To produce 30 dt of rapeseed, it is necessary to provide 213 kg of nitrogen (N), 89 kg of phosphorus ($P_2O_5$), 287 kg of potassium ($K_2O$), 157 kg of calcium (CaO), 70 kg of nitrogen (MgO) and 75 kg of sulfur (S) [38]. Of the micronutrients, rapeseed shows the greatest sensitivity to and high demand for boron and manganese. The demand according to various authors ranges from 80 g ha$^{-1}$ of boron and 100 g ha$^{-1}$ of manganese [39] to 300 g of boron and 500 g of manganese for the assumed yield of 5 t ha$^{-1}$. Regarding other micronutrients, rapeseed needs about 50–200 g of copper, 300–750 g of zinc and 5–10 g of molybdenum for the assumed yield of 5 t ha$^{-1}$ [40].

*2.4. Corn*

Corn is a crop of major economic importance worldwide [41]. The volume of yield and the acreage devoted to its cultivation place it among the three most important crops, next to wheat and rice [42]. It is also of great importance in our country, due to its multiple uses, mainly for grain and silage, but it can also be used in the production of biogas or biofuels. The last few decades have seen a significant increase in the area under cultivation. In Poland, in 1995, the species occupied an area of 181 tys. ha, while in 2009, the cultivation area increased to 695 tys. ha [43]. In 2019, the area of corn cultivated for grain was 665 tys,

and that for silage, 600 tys. ha, giving us a total of 1.26 million ha and placing it second, in the area of sown crops in Poland, after cereals and just ahead of rapeseed [35].

Table 4 summarizes the average yield of corn in Poland in the period from 2010 to 2020, with average grain yield ranging from 47.1 to 71.5 dt ha$^{-1}$, and green forage, from 357 to 493 dt ha$^{-1}$ (Table 4) [35].

**Table 4.** Corn yield in Poland.

| Use of the Corn Crop | 2010 | 2015 | 2016 | 2017 | 2018 | 2019 | 2020 |
|---|---|---|---|---|---|---|---|
| | dt ha$^{-1}$ | | | | | | |
| Grain | 59.7 | 47.1 | 72.9 | 71.5 | 59.9 | 56.2 | 56.2 |
| Forage | 437 | 357 | 493 | 487 | 426 | 406 | 459 |

Corn is characterized by a sizable demand for nitrogen; it is assumed to require 25 kg of this element to produce one ton of grain and a corresponding weight of straw [44]. It is also characterized by a high demand for phosphorus and potassium. To produce 1 ton of grain with adequate straw, it requires about 10 kg of phosphorus ($P_2O_5$) and 30 kg of potassium ($K_2O$). Thus, for the yield of 10.0 tons of grain ha$^{-1}$, the nutritional needs are 250 kg of nitrogen (N), 100 kg of phosphorus ($P_2O_5$) and 300 kg of potassium ($K_2O$). Good sulfur supply for corn plays a key role in nitrogen utilization and conversion [45], enabling higher yield with less nitrogen fertilization. The beneficial effect of sulfur on increased plant uptake and utilization of the applied nutrient has also been reported [46]. Balanced plant nutrition should take into account the supply of both macronutrients and micronutrients to plants, which affect the efficiency of the uptake of the supplied components [47] but also contribute to the increase in the obtained yield themselves [48].

Corn is very sensitive to zinc and boron deficiencies, while it is less sensitive to deficiencies of copper, manganese and molybdenum. Fertilization with micronutrients is particularly important because corn is mainly grown on light soils, which are characterized by a much lower content of micronutrients compared with heavy soils. The requirement for corn to produce one ton of grain in relation to micronutrients is as follows: 20 g of boron (B), 12 g of copper (Cu), 45 g of iron (Fe), 35 g of manganese (Mn), 1 g of molybdenum (Mo) and 50 g of zinc (Zn). The use of micronutrients in corn production contributes to higher yield, but the effectiveness of micronutrient fertilization treatment depends, among other things, on the availability of macronutrients [48], indicating that the combined application of macro- and micronutrients produces better yield-forming effects [49].

*2.5. Types of Foliar Fertilizers*

The number of foliar fertilizers available on the Polish market has been steadily increasing for several years, mainly due to the growing demand of agricultural producers for dedicated, specialized products adapted to the requirements of individual crops. Scientific research confirming the applicability of the products in agricultural practice contributes to the increase. We can divide the available products into several different groups, depending on the selected criterion:

1. Physical form: (a) liquid fertilizers—in this group, we can distinguish among liquid fertilizers, fertilizers in suspension and gel; (b) loose fertilizers—soluble in water, they are in the form of powders and crystals of various shapes.

2. Purpose: (a) universal—fertilizers that can be used in any crop; (b) dedicated— tailored to the nutritional needs of selected plants and crops.

3. Amount of components in the fertilizer: (a) monocomponent—dominant, high content of one macro- or micronutrient (N, B, Cu, Zn, Mn); (b) bicomponent—high content of two components, whether macronutrients (N + Ca, P + K, S + Mg, N + K, etc.), micronutrients (B + Mo, B + Zn, B + Mn, Mn + Zn) or a mix of macro- and micronutrients (N + Mn, N + Mo, P + B, P + Zn); (c) multicomponent—containing a minimum of three or more nutrients (N + P + K, N + P + K+ micro, P + K + Mg, Zn + B + Mg).

4. Type of components: (a) primary macronutrients with high content—N, P, K; (b) secondary macronutrients with high content—Ca, Mg, Na, S; (c) micronutrients with high content—B, Cu, Fe, Zn, Mn, Mo; (d) mixed fertilizers—fertilizer mixtures with increased content of selected macro- and micronutrients.

5. Forms of nutrients: (a) "pure" ionic forms; (b) complexed—with complexing agents, such as lignosulfonic acid (LS), glutamine hydroxamate (HGA), organic acids; (c) chelated—chelating agents, such as ethylenediaminetetraacetic acid (EDTA), 2-hydroxyethylethylenediaminetriacetic acid (HEEDTA), diethylenetriaminepentaacetic acid (DTPA), ethylenediamine-N,N′-bis(2-hydroxyphenylacetic) acid (EDDHA), ethylenediamine-di (o-hydroxy-o-methylphenylacetic) acid (EDDHMA), ethylenediamine di-(2-carboxy-5-hydroxyphenylacetic) acid (EDDCHA), ethylenediamine-N-N=bis(2-hydroxy-5-sulfophenylacetic) acid (EDDHSA), N-(1,2-dicarboxyethyl)-D,L-aspartic acid (IDHA), N,N-di(2-hydroxybenzyl)ethylenediamine-N,N-diacetic acid (HBED) and ethylenediaminedisuccinic acid (EDDS).

6. Homogeneity: (a) complex fertilizers—fertilizers containing at least two nutrients, characterized by the fact that in the solid phase, each granule has exactly the same content of each of the declared components; (b) blended fertilizers (blends)—fertilizers resulting from the physical mixing of at least two other fertilizers, without chemical reactions.

7. The content of additional components: (a) deficient components—silicon (Si), iodine (I), chromium (Cr); (b) adjuvants—supportive agents, improving the effectiveness of foliar fertilization treatments; (c) anti-stress substances—substances that increase resistance to stress factors(amino acids, plant extracts, vitamins); (d) stimulants—substances and chemical compounds that stimulate the plant for intensive development (e.g., amino acids, hormones, humic substances).

*2.6. Foliar Fertilizer Market*

The foliar fertilizer market in Poland is currently estimated at around PLN 300 million. The main suppliers of the products are Polish companies, but many solutions from Europe and further corners of the world are also available. We can count, among the leaders of the domestic market, the companies ADOB®, EKOPLON® and INTERMAG®. These are companies that have been engaged in the production of foliar fertilizers for more than 30 years, supplying their products both to the Polish market and to many other countries around the world. Among foreign manufacturers, the Polish market is dominated by European companies coming from Italy, France, Spain, Great Britain, Belgium and Turkey, but there are also companies from the United States of America, China or Japan. Every year, new domestic and foreign manufacturers of foliar fertilizers and biostimulants appear on the Polish market. Such a large number of suppliers allows Polish agricultural producers to benefit from the latest solutions in the field of foliar fertilization and biostimulation.

The market of foliar fertilizers has undergone quite a transformation over the past 30 years. In the 1990s, liquid fertilizers containing basic macro- and micronutrients dominated the market. Some of these products are still available on the market and do not promise to disappear in the near future. An example of such a product is liquid boron fertilizers containing boron in organic form—boroethanolamine—which account for about 70–80% of all boron fertilizers used on farms. The remaining 20–30% is in bulk, soluble forms based on boric acid and sodium borates. Blends of boron fertilizers with macro- and micronutrients are also available.

Between 2000 and 2010, loose soluble fertilizers began to appear and gain importance on the market. These were products mainly containing nitrogen, phosphorus and potassium, along with micronutrients. Four products were standard in the offer of manufacturers: balanced fertilizer, containing nitrogen, phosphorus, potassium and micronutrients at the same level; fertilizer with increased nitrogen content; fertilizer with increased phosphorus content; and fertilizer with increased potassium content. Compared with liquid fertilizers, bulk fertilizers require slightly longer preparation time due to the need for pre-dissolution. However, in many cases, they have a much higher content and concentration of nutri-

ents. In addition, they are cheaper to transport and more resistant to changing weather conditions during storage.

The biggest boom in the market of foliar fertilizers occurred after 2010, at which time, macro- and micronutrient products, enriched with various additives, began to appear. Currently, on the market, we have many innovative solutions tailored to the current specifics of production, plant requirements and changing environmental conditions.

In the tables below (Tables 5–7) is a comparison of selected fertilizers dedicated to the cultivation of cereals, rapeseed and corn, as well as popular directions in which producers and science are directing their research and development efforts.

**Table 5.** Selected foliar fertilizers used in cereal crops.

| Foliar Fertilizer | Chemical Composition | | | | | | | | | | | Type of Complexes/Chelates |
|---|---|---|---|---|---|---|---|---|---|---|---|---|
| | N | P$_2$O$_5$ | K$_2$O | MgO | SO$_3$ | B | Cu | Fe | Mn | Mo | Zn | |
| Liquid fertilizers | | | | | | | | | | | | |
| Ekolist cereals | 127 | 38.1 | 38.1 | - | - | 0.13 | 11.4 | 8.9 | 10.1 | 0.06 | 8.9 | EDTA organic complexes |
| Plonvit cereals | 195 | - | - | 26 | 59 | 0.18 | 11.7 | 10.4 | 14.3 | 0.06 | 13 | EDTA organic complexes |
| Vital cereals | 198 | - | - | 66 | - | 1.32 | 3.3 | 6.6 | 11.8 | 0.16 | 13.2 | No |
| Sarplon cereals | 245 | - | 34.3 | 10 | - | 1.32 | 6.6 | 13.2 | 19.8 | 0.26 | 4 | EDTA/DTPA |
| Suplofol Micro Z | 204 | - | - | 27 | 68 | 2 | 6.8 | 13.6 | 25.9 | 0.2 | 13.6 | No |
| Crystalline fertilizers | | | | | | | | | | | | |
| Maximus amino micro cereals | 11 | - | 70 | - | - | 3.4 | 50 | 20 | 40 | 0.4 | 20 | Glicyna |
| Adob Micro cereals | 100 | - | 50 | - | 310 | - | 15 | 3 | 30 | 0.2 | 5 | EDTA |
| Amino Ultra cereals | - | - | - | 20 | - | 1.6 | 16 | 65 | 65 | 0.7 | 49 | Glicyna |
| Cereals forte | 50 | 150 | 150 | 78 | 20 | 0.2 | 10 | 1 | 10 | 0.01 | 0.04 | EDTA/DTPA |
| Dr Green cereals | - | - | - | - | - | 5 | 50 | 60 | 80 | 0.5 | 20 | Micro Activ |
| Opti cereals | 140 | 160 | 160 | 30 | 180 | - | 3 | 1.5 | 5 | 0.4 | 1.5 | EDTA/DTPA |
| Suspension fertilizers | | | | | | | | | | | | |
| Yaravita Gramitrel | 64 | - | - | 250 | - | - | 50 | - | 150 | - | 80 | Oxide form |

**Table 6.** Selected foliar fertilizers used in rapeseed cultivation.

| Foliar Fertilizer | Chemical Composition | | | | | | | | | | | Type of Complexes/Chelates |
|---|---|---|---|---|---|---|---|---|---|---|---|---|
| | N | P$_2$O$_5$ | K$_2$O | MgO | SO$_3$ | B | Cu | Fe | Mn | Mo | Zn | |
| Liquid fertilizers | | | | | | | | | | | | |
| Ekolist rape | 150 | 50 | 37.5 | - | - | 8.7 | 0.1 | 8.7 | 8.7 | 0.06 | 0.12 | Technology ACTIVE |
| Plonvit rape | 186 | - | - | 31 | 31 | 6.2 | 1.2 | 6.2 | 6.2 | 0.06 | 6.2 | Technology INT |
| Vital rape | 188 | - | - | 40 | - | 6.9 | 1.2 | 3.1 | 4.4 | 0.09 | 3.7 | No |
| Sarplon rape | 285 | - | 30.4 | 12.4 | 1.45 | 5.3 | 2.38 | 2.38 | 16.8 | 0.53 | 2.4 | EDTA/DTPA |
| Suplofol micro BR | 195 | - | - | 26 | 65 | 6.2 | 0.85 | 1 | 23.4 | 0.2 | 9.5 | No |

**Table 6.** *Cont.*

| Foliar Fertilizer | Chemical Composition | | | | | | | | | | | Type of Complexes/Chelates |
|---|---|---|---|---|---|---|---|---|---|---|---|---|
| | N | $P_2O_5$ | $K_2O$ | MgO | $SO_3$ | B | Cu | Fe | Mn | Mo | Zn | |
| Crystalline fertilizers | | | | | | | | | | | | |
| Maximus Amino Micro rape | 110 | - | 70 | - | - | 20 | 15 | 30 | 40 | 0.4 | 15 | Glicyna |
| Adob Micro rape | 47 | - | - | - | 135 | 100 | 5 | 3 | 15 | 1 | 3 | EDTA |
| Rapeforte | 50 | 150 | 150 | 46 | 147 | 30 | 0.03 | 1.5 | 10 | 0.01 | 0.04 | EDTA/DTPA |
| Dr Green rape | - | - | - | - | - | 100 | 2 | 25 | 50 | 0.5 | 20 | Micro Activ |
| OPTI rape | 110 | 150 | 210 | 20 | 190 | 15 | 1 | 1.5 | 2 | 0.4 | 1.5 | EDTA |
| Suspension fertilizers | | | | | | | | | | | | |
| Yaravita Gramitrel | 64 | - | - | 250 | - | - | 50 | - | 150 | - | 80 | Oxide form |

**Table 7.** Selected foliar fertilizers used in corn cultivation.

| Foliar Fertilizer | Chemical Composition | | | | | | | | | | | Type of Complexes/Chelates |
|---|---|---|---|---|---|---|---|---|---|---|---|---|
| | N | $P_2O_5$ | $K_2O$ | MgO | $SO_3$ | B | Cu | Fe | Mn | Mo | Zn | |
| Liquid fertilizers | | | | | | | | | | | | |
| Ekolist corn | 75.6 | 126 | 37.8 | - | - | 6.3 | 1.2 | 7.5 | 2.5 | 0.06 | 11.3 | Technologia ACTIVE |
| Plonvit corn | 195 | - | - | 26 | 54.6 | 5.2 | 7.8 | 9.1 | 9.1 | 0.065 | 14.3 | Technologia INT |
| Vital corn | 203 | - | - | 69 | - | 1.6 | 4.1 | 6.7 | 13.5 | 0.13 | 17.6 | Brak |
| Sarplon corn | 271 | - | 11.9 | 18.5 | - | 4 | 1.3 | 1.3 | 6.6 | 0.46 | 19.8 | EDTA |
| Suplofol Micro K | 188 | - | - | 25 | 63 | 2.5 | 1.25 | 3.75 | 5 | 0.38 | 20 | No |
| Crystalline fertilizers | | | | | | | | | | | | |
| Maximus Amino Micro corn | - | 110 | 70 | - | - | 20 | 20 | 20 | 30 | 0.4 | 50 | Glicyna |
| Adob Mikro corn | 70 | 20 | - | 30 | 100 | 20 | 1 | 2 | 5 | 0.1 | 40 | EDTA |
| Corn forte | 50 | 200 | 150 | 42 | 220 | 15 | 0.07 | 1 | 0.1 | 0.01 | 15 | EDTA. DTPA |
| Dr Green corn | - | - | - | - | - | 5 | 2 | 60 | 70 | 0.5 | 80 | Micro Activ |
| OPTI corn | 100 | 210 | 140 | 30 | 140 | 5 | 2 | 1 | 0.3 | 3 | 1 | EDTA |
| Suspension fertilizers | | | | | | | | | | | | |
| YaraVita Zeatrel | - | 440 | 75 | 67 | - | - | - | - | - | - | 46 | Oxide form |

From the compilation of fertilizers in the tables (Tables 5–7), each product, even if dedicated to the same crop, is characterized by a different composition. On the market, we have products that contain all basic macro- and micronutrients, as well as those that only contain selected nutrients. Individual products also differ in the content of additives, the main purpose of which is to improve the efficiency of fertilization and/or support the plant in case of biotic and abiotic stresses. Large discrepancies can also be seen in the recommended dosages and prices of individual fertilizers. The above comparison shows that when choosing a particular solution, many factors should be taken into account, including composition, forms of components and their availability to the plant, as well as dosage, the price of the fertilizer or the type of chelating, complexing substances, which should be selected to maximize the effectiveness of foliar fertilization under specific conditions in the field.

### 2.7. Foliar Fertilizers Containing Amino Acids

The last 10 years have seen a significant increase in interest in amino acids used in plant fertilization. Currently, on the market, we have many products containing amino acids. Depending on the product, the content of these components can vary from less than 1% to more than 40% of the total weight of the fertilizer [23]. The various solutions also differ in the source of amino acids, e.g., plant or animal origin, and in the content of individual amino acids. All amino acids except glycine can differ in optical activity. They can exist in L- and D-forms, and very importantly, both forms can be taken up and utilized by plants [50]. There have been many publications on the effectiveness of amino acids, with authors confirming the effectiveness of fertilizers containing amino acids of both plant and animal origin [51]. The effectiveness of amino acids is due, among other things, to their nitrogen content, which is necessary for optimal plant growth and development, as well as their effect on the efficiency of uptake and utilization of other nutrients [23]. They can also influence the content of chlorophyll or carotenoids in plants, key substances involved in photosynthesis [52,53].

### 2.8. Foliar Fertilizers Containing Nano-Elements

Among the products used in foliar fertilization are those that contain nanocomponents. Their effectiveness can depend on the timing of application, concentration or particle size [54]. These fertilizers are gaining importance due to their ability to reduce the negative impact of fertilizers on the environment and their effectiveness. A positive effect of nano-iron on fruit yield and quality parameters was shown by [55], while in an experiment with rapeseed, a yield increase of 1298 kg ha$^{-1}$ was reported [56]. Vishekaii et al. [57] studied the effect of boron nano-chelates on fruit and olive oil yield. Among the available literature, one can also find information on the positive effects on growth and yield of nano-molybdenum [58], nano-zinc [54] or nano-silicon [59], nano-copper, and nano-silver [60]. Meena et al. [61] studied the effects of nano-phosphorus, potassium and zinc on wheat cultivation, showing positive effects of nanoparticles on growth and yield at levels ranging from a few to tens of percentage points, depending on the combination.

### 2.9. Foliar Fertilizers Containing Silicon

For several years, the foliar fertilizer market has seen an increase in interest in foliar fertilizers containing silicon. Among other things, the use of these products contributes to the strengthening of plant cell walls, thereby increasing resistance to damage [62]. Foliar application of silicon also has a positive effect on reducing water loss through the leaves [63]. The positive effect of silicon on the response of plants to high temperatures was demonstrated in their experiment by Basirat and Mousavi [64]. The experiment was conducted in a greenhouse under controlled, high-temperature conditions (36 °C). Foliar application of silicon resulted in a 36.1% increase in total cucumber yield and a 40.3% increase in marketable yield. The positive effect of silicon applied in the form of nanoparticles was also demonstrated by Shalaby et al. [59]. In their study, they showed that the effectiveness of silicon fertilizers depended on the number of fertilization treatments and habitat conditions, and that yield gains could range from 6.4% (one treatment) to 12.9% (three treatments) in years with optimal rainfall and from 12.2% to 17.6% in dry years [63].

### 2.10. Foliar Fertilizers for Biofortification

The word biofortification is becoming more and more popular every year. There is more and more talk about deficiencies of selected elements in the diet of humans and animals and thus the need to develop effective methods for providing these elements [65]. One of the most effective ways is plant biofortification, that is, enriching plants with specific elements and improving their availability. The process of biofortification can take place via fertilization (topdressing and foliar) or breeding varieties that are able to accumulate increased amounts of selected elements. Foliar fertilization is one of the most popular and effective methods of plant biofortification, as it allows the delivery of the right amount of

specific elements in a specific developmental stage of the plant and is thus an economical and environmentally safe procedure.

Most often, iodine and selenium are mentioned in biofortification, but the topic can also apply to other nutrients. In the case of iodine, the minimum human requirement is 1 µg kg$^{-1}$ body weight day$^{-1}$, and the optimal dose is 95–150 µg day$^{-1}$. The human body has a similar requirement for selenium, where the optimal level for one person is 50–200 µg day$^{-1}$ [19]. Currently, there are no foliar fertilizers for biofortification on the domestic market, but scientific research has been conducted for more than 10 years to determine the optimal doses and chemical forms of iodine or selenium. The effects of foliar and soil application of iodine and selenium in their experiments were studied by [66–69]. Biofortification using agrotechnical methods is a direction that will become increasingly important in the coming years, as the problem of nutrient deficiency in the diet of humans and animals affects not only our country but also the entire globe.

## 3. Conclusions

The analysis of the foliar fertilizer market in Poland indicates that its development is very dynamic. High competition among manufacturers of foliar fertilizers contributes to the appearance of many new innovative products on the market. Currently, products enriched with natural or synthetic additives to improve the efficiency of fertilization are expected to become the standard.

**Author Contributions:** Conceptualization, R.J., G.K. and M.S., data curation—compilation and analysis of the results, R.J. and G.K.; writing–original draft preparation, R.J. and G.K.; writing–review and editing, R.J., G.K. and M.S. All authors have read and agreed to the published version of the manuscript.

**Funding:** The APC/BPC is financed/co-financed by Wrocław University of Environmental and Life Sciences.

**Institutional Review Board Statement:** Not applicable.

**Data Availability Statement:** Not applicable.

**Conflicts of Interest:** The authors declare that they have no conflict of interest.

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
