# Peer review of "Foliar Fertilization of Crop Plants in Polish Agriculture"

_agriculture, doi:10.3390/agriculture13091715_

Round 1

Reviewer 1 Report

The authors analyzed the market for foliar fertilizers in Poland, which developed dynamically in recent years. The topic is of interest, the manuscript is well written in the sense that it is clear and understandable, however, it could be improved by considering the comments and suggestions below:

1. The folia fertilizer market in Poland has been analyzed, so should the title of the manuscript be limited to Poland, as “Foliar fertilization of crop plants in Poland—a comprehensive analysis”?

2. Is it possible for the authors to add some content about the regulatory requirements for foliar fertilizer entering the market, especially fertilizer containing nano-elements, and are there any special marketing regulations and restrictions for nano-products in Poland?

3. Abbreviations should follow their full names when they first appear (PLN, HGA, EDTA, HEEDTA, DTPA, EDDHA, EDDHMA, EDDCHA, EDDHSA, IDHA, HBED, EDDS…).

Some statements in the manuscript need to be refined, kindly remind the authors to check the whole manuscript.

Line 19 “ is one of the more popular….”, should be “one of the most popular…..”

Line 151 and 180, per 1 hectare, and per 1 ha, the number “1” should be deleted and it’s better the unit of land area be consistent throughout the text.

Author Response

Dear Reviewer,

Thank you very much for your very constructive and valuable opinions and suggestions that will improve our manuscript. Thank for your times and engagement.

Preparing the current version of manuscript we took into account almost all suggestions and recommendations.

Response to Reviewer 1 Comments

Below is presented general information about  a correction that  was done:

  1. The folia fertilizer market in Poland has been analysed, so should the title of the manuscript be limited to Poland, as “Foliar fertilization of crop plants in Poland— a comprehensive analysis”?

The title of the article has been changed and clarified

"Foliar fertilization of crop plants in the polish agriculture"

  1. Is it possible for the authors to add some content about the regulatory requirements for foliar fertilizer entering the market, especially fertilizer containing nano-elements, and are there any special marketing regulations and restrictions for nano-products in Poland?

According to the regulation, in order for a fertilizer to be approved for marketing, it must be made from raw materials that meet the requirements of one of the component material categories (CMC) in Annex 2 [27]. So, among other things, what is important is the ingredient itself, its origin or its purity, while it is not important, for example, the particle size or chemical form of the element in concern. In practice, most manufac-turers aim to introduce and make their solutions available in accordance with the EU regulation (mainly due to the freedom of circulation in member countries), and if this is not possible, then national regulations are used.

  1. Abbreviations should follow their full names when they first appear (PLN, HGA, EDTA, HEEDTA, DTPA, EDDHA, EDDHMA,EDDCHA, EDDHSA, IDHA, HBED, EDDS…).

Added full names before abbreviations when they first appear

  1. Line 19 “ is one of the more popular….”, should be “one of the most popular…..”

Modified

  1. Line 151 and 180, per 1 hectare, and per 1 ha, the number “1”should be deleted and it’s better the unit of land area be consistent throughout the text.

Modified

Reviewer 2 Report

The paper is well written needs moderate modification for final submission

English of the paper is well written needs moderate modification for final submission

Author Response

Dear Reviewer,

Thank you very much for your very constructive and valuable opinions and suggestions that will improve our manuscript. Thank for your times and engagement.

Preparing the current version of manuscript we took into account almost all suggestions and recommendations.

Response to Reviewer 2 Comments

Below is presented general information about  a correction that  was done:

  • Mechanism of nutrient uptake through the foliar parts should added along with diagram for better understanding.

Added a short paragraph on the subject in Introduction.

The delivery of nutrients with foliar fertilizers generally involves: 1) the application of an aqueous solution to the plant surface in the form of droplets 2) their retention on the leaf surface 3) absorption of the nutrient into living plant cells 4) translocation of the nutrient to areas where it can be used by the plant in its life processes. Efficiency of foliar fertilization depends largely on the absorption mechanisms of foliar-applied molecules, as they are affected by many environmental factors. There are re-view papers in the literature that focus on the mechanisms of penetration of foliar-applied nutrient solutions through the leaf surface [8–10]

2) Reframe the para for better understanding of the readers

Rewritten text on lines 49-54.

Achieving these goals is possible, but it means developing more efficient and effective fertilizer systems. Their goal should be to increase agricultural productivity with less fertilizer inputs [6]. Of the fertilization methods currently in use, such expectations are largely met by foliar delivery of nutrients, the use of which is up to 30 times greater than with soil application [7].

  • Area (name of the country) should be clearly mentioned

It was specified that this is about Poland.

  • graphical/ tabular representation of data should be given
  • Diagrammatic representation should be given for better understanding

In my opinion, it is sufficient to provide this information in text form.

  • Should be mor consisted focusing the main points

Rewritten and shortened conclusions.

Analysis of the foliar fertilizer market in Poland indicates that its development is very dynamic. High competition among manufacturers of foliar fertilizers contributes to the appearance of many new innovative products on the market. Currently, products enriched with natural or synthetic additives to improve the efficiency of fertilization will become the standard.